# Computer Vision and IoT-Based Sensors in Flood Monitoring and Mapping: A Systematic Review

**DOI:** 10.3390/s19225012

**Published:** 2019-11-16

**Authors:** Bilal Arshad, Robert Ogie, Johan Barthelemy, Biswajeet Pradhan, Nicolas Verstaevel, Pascal Perez

**Affiliations:** 1SMART Infrastructure Facility, University of Wollongong, Wollongong 2522, NSW, Australia; rogie@uow.edu.au (R.O.); johan@uow.edu.au (J.B.); pascal@uow.edu.au (P.P.); 2The Centre for Advanced Modelling and Geospatial Information Systems (CAMGIS), Faculty of Engineering and Information Technology, University of Technology Sydney, Sydney 2007, NSW, Australia; Biswajeet.Pradhan@uts.edu.au; 3Department of Energy and Mineral Resources Engineering, Sejong University, Choongmu-gwan, 209 Neungdong-ro, Gwangjingu, Seoul 05006, Korea; 4UMR 5505 CNRS-IRIT, Université Toulouse 1 Capitole, 31062 Toulouse, France; nicolas.verstaevel@irit.fr

**Keywords:** remote sensing, flood, disaster management, coastal, environmental sensor network (ESN), IoT, drones, UAV, computer vision, wireless sensor network

## Abstract

Floods are amongst the most common and devastating of all natural hazards. The alarming number of flood-related deaths and financial losses suffered annually across the world call for improved response to flood risks. Interestingly, the last decade has presented great opportunities with a series of scholarly activities exploring how camera images and wireless sensor data from Internet-of-Things (IoT) networks can improve flood management. This paper presents a systematic review of the literature regarding IoT-based sensors and computer vision applications in flood monitoring and mapping. The paper contributes by highlighting the main computer vision techniques and IoT sensor approaches utilised in the literature for real-time flood monitoring, flood modelling, mapping and early warning systems including the estimation of water level. The paper further contributes by providing recommendations for future research. In particular, the study recommends ways in which computer vision and IoT sensor techniques can be harnessed to better monitor and manage coastal lagoons—an aspect that is under-explored in the literature.

## 1. Introduction

Natural hazards such as floods, storms, tsunamis and others pose a significant threat to lives and property around the world [1]. Without proper monitoring and effective mitigation measures, these natural perils often culminate in disasters that have severe implications in terms of economic loss, social disruptions, and damage to the urban environment [2,3]. Historical records have shown that flood is the most frequent natural hazard (see Figure 1), accounting for 41% of all natural perils that occurred globally in the last decade [4]. In this period alone (2009 to 2019), there were over 1566 flood occurrences affecting 0.754 billion people around the world with 51,002 deaths recorded and damage estimated at $371.8 billion [4]. Put in context, these statistics only account for “reported” cases of large-scale floods, typically considered flood disasters. A flood disaster is defined as a flood that significantly disrupts or interferes with human and societal activity, whereas a flood is the presence of water in areas that are usually dry [5,6]. The global impact of a flood would be more alarming if these statistics incorporated other numerous small-scale floods where less than 10 people may have died, 100 or more people may have been affected or where there is no declaration of a state of emergency or a call for international assistance. Nevertheless, the current situation calls for improved ways of monitoring and responding to floods. The importance of improved flood monitoring cannot be overemphasized given the growing uncertainty associated with climate change and the increasing numbers of people living in flood-prone areas [7].

Significant efforts have been made globally to develop cost-effective and robust flood monitoring solutions. A common approach is based on computer vision, wherein relevant images from existing urban surveillance cameras are captured and processed to improve decision making about floods [8]. These types of camera-based applications involve low equipment cost and wide aerial coverage thereby enabling the detection of flood levels at multiple points. The wider coverage gives the computer vision approach an advantage over the traditional flood monitoring method that relies on fixed-point sensors [9]. Computer vision is based on image processing techniques that have been widely applied in many fields, including aerospace, medicine, traffic monitoring, and environmental object analysis [10]. In the last decade, research efforts have intensified in exploring computer vision to improve flood monitoring, flood inundation mapping, debris flow estimation, and post-flood damage estimation. To effectively harness this knowledge and foster rapid research progress, it is important to review the relevant literature and provide a constructively critical appraisal of scientific production, including recommended directions for future research.

Another method of flood monitoring and prediction is the use of wireless sensors powered by the Internet of Things (IoT) technology. IoT and computational models such as artificial neural network (ANNs) [11] have opened up new doorways, allowing the design of new hardware and software to provide real-time water-level data as required for flood monitoring and forecasting [12]. Today, many flood-prone countries, including the tropical nation of Indonesia that suffers from annual monsoonal rainfall, are exploring IoT sensors to gather intelligence for issuing early warnings and evacuate orders to people at risk of major floods [13]. The IoT has gained increased popularity in the last decade, particularly within the context of smart city applications such as real-time monitoring of urban drainage networks using wireless sensors [14]. A review of the relevant literature is needed to provide an in-depth understanding of the research scope and progress achieved in the last decade of using IoT sensors for flood monitoring in both occupied lands and other coastal sites such as lakes and lagoons.

This study provides an opportunity to update readers on recent advancements in flood monitoring, and how technology is used in the literature to map the flood events. The motivation behind this study is to highlight existing solutions and adapt them to better manage coastal lagoons, which impose flood threat to the local communities. This study presents a systematic review of the literature focusing on the use of computer vision and IoT-based sensors in flood monitoring, mapping and prediction for both occupied lands and coastal sites such as lagoons. The main contributions of this article are as follows:A detailed survey is presented on the use of computer vision and IoT-based sensors for flood monitoring, prediction and inundation mapping. The scope covers the state-of-the-art applications of computer vision and sensor integrated approaches for managing coastal sites and other flood-prone urban areas.The study highlights gaps in the literature and recommends directions for future research.

The following section presents the methodology adopted in conducting this systematic literature review.

## 2. Methodology

This section provides details for the procedure involved in the selection, inclusion and exclusion of research articles. The review was conducted using the Preferred Reporting Items for Systematic Reviews and Meta-Analysis (PRISMA) guidelines [15]. Overall, three databases were selected to conduct this review, namely, Scopus, IEEE Xplore and Science Direct. The keywords utilised to select relevant articles from the databases are listed in Table 1 along with the number of retrieved research papers.

The research articles were manually screened by reading the title and abstract. The database search returned (*n* = 13,875) records from three online databases. After removal of duplicate articles, only 2823 articles were left for review. The titles and abstracts of these 2823 articles were manually screened for relevance, resulting in the exclusion of 2415 records. The remaining 408 articles were selected for full-text review and content analysis. For inclusion in the final list, articles were required to be published between 2009 and 2019 and to be related to flood monitoring, forecasting or mapping. These inclusion criteria resulted in 91 relevant articles. In regard to exclusion criteria, the articles about IoT protocols in flood monitoring were not included in this review, as this is not the core focus of this study. Furthermore, duplicated articles in different databases were also discarded, and only the articles written in English were considered in this review. The PRISMA flow diagram for the systematic literature review can be seen in Figure 2.

## 3. Computer Vision and IoT Sensors for Early Warning Systems

Remote sensing technologies, such as computer vision and wireless sensor networks (WSNs), are increasingly used in the literature to support early warning systems [16]. An early warning system provides advanced warnings in case the water level is likely to rise and reach the alarming flood level. These systems can generate notifications via SMS alerts, emails or through a web server. An early warning system can, for example, help to send alerts or warnings to local occupants and motorists so that they can avoid the usage of flooded roads. This section will focus on studies that have utilized computer vision and IoT-based sensors for improving early-warning initiatives. This discussion covers several research areas that are useful for supporting early warning systems, including the estimation of water levels through camera images, IoT-based sensor approaches for water level estimation and the use of computer vision for the collection of flood-related data.

### 3.1. Computer Vision for Estimating the Water Level

The monitoring of water levels is of extreme importance in early warning systems, and computer vision has shown to be useful [17]. Image filtration in computer vision plays a vital role in estimating water levels [18]. For example, Yu et al. [18] proposed the differencing image technique to track and detect minor changes in water level. The difference method is based on analysing the region of interest (ROI) between the previous and current frame and then outputting a level of water by using the Otsu threshold method. The acquired image from the river is first filtered by using a Gaussian and averaging filter that helps to minimize the noise. The water level is then estimated from the *y*-axis of the edged image. The experiment was performed in only one location. Given that a threshold for the different filters will change under different illuminations, it will be interesting to investigate the robustness of this approach by conducting the experiment in different locations. A similar approach to the differencing technique has been proposed by Hiroi et al. [19]. The proposed remote sensing solution also utilises the differencing technique to observe water levels via cameras.

However, this approach involves taking images at 10 min intervals, comparing every successive image with the last reading and then estimating the water level by using logistic regression. The solution was successfully tested on 13 different locations, reliably predicting a rise in water level with decent accuracy.

Another study utilised the physical measuring ruler along with different computational models in computer vision, including the differencing method, dictionary learning and convolutional neural network (CNNs) [20]. The dictionary method is based on classifying the ROI into two classes, i.e., ruler and water region. The features of the water and ruler are stored in the dictionary. By analysing the boundary line between the ruler and water classes, the water level can be calculated. The CNN delivered the most promising results. A CNN is a computer vision technique which involves convolving the image with the filter. The role of the filter is to extract important features from the image. The algorithm was trained on raw images and during prediction. Instead of using preserved features from the dictionary, the algorithm extracted features from the input image. Having tested the algorithms on six different locations, the study concluded that the CNN outperformed the accuracy of both the dictionary learning technique and the differencing method. The average error and variance of error recorded for the three different methods can be seen in Table 2.

The task of computationally differentiating a water body in an image can be challenging. A vital step is to rely on the intensity data from the water body to develop a mathematical model that contains the water body reflection coefficients [21]. Rankin et al. [22] considered the low texture part of the image as the water body. Low texture in an image can be found by converting the red/green/blue (RGB) image to grayscale and convolving a grayscale image with a 5 × 5 intensity variance filter. The study utilized the intensity data from the water body to extract the reflection coefficient from surface reflection. In contrast to using only intensity information, Park et al. [23] proposed the segmentation technique to identify the water level. The proposed algorithm uses an accumulated histogram approach and a bandpass filter. The bandpass filter is fine-tuned to reduce the noise in the image. For this reason, images taken from a charge-coupled device (CCD) camera are converted from time series to a frequency domain using discrete cosine transform (DCT). In the accumulated histogram approach, the image is compared with previous frames and a histogram plotted, so that the changes in the histogram can be tracked, and the water level estimated from variation in the histogram. In a similar approach, Udomsiri et al. [24] proposed the edge detector finite impulse response (FIR) filters along with bandpass filter to find the boundary between water and ground. The water level was detected by finding features of horizontal straight lines. The error of the detection was calculated by measuring the water level manually and comparing the results with the output of the algorithm.

Moreover, Zhang et al. [25] has proposed a real-time flow and water level measurement system based on near infrared (NIR) imaging, OSF-based adaptive thresholding and image ortho-rectification techniques. The proposed framework consists of ten steps as follows: (i) camera calibration to obtain intrinsic and distortion matrix; (ii) correction of non-linear distortion of an image; (iii) selection of staff gauge ruler as ROI; (iv) design of binary orthographic template image based on chosen ROI; (v) selection of corresponding points on staff gauge; (vi) determining the transformation matrix of a staff gauge with respect to the camera; (vii) ortho-rectification of ROI image; (viii) segmentation of an image with adaptive thresholding; and (ix) locating the water line in the image by accumulating grey values in a row.

In contrast to utilizing visual information based only on ground or wall-mounted cameras [26], Ridolfi et al. [27] deployed unmanned aerial vehicles (UAVs) to monitor the water level in dams. The water level was estimated by utilizing a canny filter on greyscale images. The threshold parameters minimum and maximum (0.019,0.047) were predefined, and the objective was to draw a boundary between water and surface. By comparing the water level retrieved from the images with a benchmarked value obtained from a traditional device, the method was found to have achieved 0.05 m in the overall mean error between the estimated and actual water levels. This outcome is quite encouraging, considering that testing in four different locations within an Italian artificial lake has reaffirmed the reliability of the method for extracting the water level from images. Image parsing is another key challenge in the use of computer vision for flood monitoring. Lo et al. [28] designed an image parser to analyse images that have significant perceptual recognizability. Firstly, the image parser looks for dark sample pixels or blank images, where the intensity of the pixel is the luminance in the hue, saturation and value (HSV) colour domain. Images with an intensity less than a specified threshold are discarded. The second step is to check the image visibility by calculating the overall luminance of an image. Afterwards, the next step is to draw some reference sampling points on an image, check the visibility at those points and then check the presence of fog/haze on site. The final phase involves checking for the presence of water in the ROI by finding geometric boundaries and edges in the resultant image.

One of the most significant findings to emerge for this subsection is that a computer vision approach can be used to extract the water level at multiple points within a field of view (FOV) of the camera. The water level readings can be validated by analysing the visual data acquired from the visual sensor. This provides an inexpensive way to forecast flood by merely relying on remote sensing data. This has also contributed to the understanding of how different computer vision methods are used in the literature.

### 3.2. IoT-Based Sensors for Estimating Water Level

There are several available sensors which are useful for estimating water level and, thus, improving early warning systems [29]. Bączyk et al. [30] discussed the pros and cons of using these sensors to monitor and measure water level. The first type of sensor is a pressure transducer. Automatic pressure transducers can measure up to some 0.001 m accuracy in water level and are compatible with many of the controllers for logging data or visualising data in real time. On the downside, automatic pressure transducers require calibration and are highly sensitive to any vertical displacement from the point of installation, as this can potentially degrade the accuracy of water level measurement as a result of changes in hydrostatic pressure [29]. Additional sensors may be required for air pressure monitoring to adjust the output of pressure transducers. Rangefinder sensors can be a great option, but these devices are often non-submersible. Rangefinder sensors are low-cost devices which makes them affordable, particularly when several sensor nodes are required to monitor a large area. However, rangefinder sensors also require manual calibration and are dependent on the distance from the measurable water level. In that sense, rangefinder sensors are popular when it comes to finding the distance from an object [31]. Essentially, ultrasonic/rangefinder sensors transmit a signal and calculate the time in between the send and receive signals as in the case of water level monitoring [32].

Similarly, optical and radar sensors play a vital role in flood monitoring and assessment through satellites [33]. Lin et al. [34] made a comparison between optical and radar sensors. The study indicated that the data from the optical sensor is widely available and Landsat is the most popular source of data extraction. However, optical sensors cannot penetrate through clouds, whereas radar uses microwaves and can penetrate through clouds. Nevertheless, optical sensors are more popular in image acquisition for scientific research because the overall cost of data collection and interpretation is not as expensive as radar. The study concluded that the best result of flood assessment is possibly by combining both optical and radar data sources. However, the European Space Agency (ESA) now provides satellite radar data from Sentinel 1a and 1b at no charge for research activity. Moreover, Khan et al. [35] proposed a novel technique to forecast flash floods by observing the increase in the level of soil moisture and carbon dioxide sensors. The research indicates that rising readings of both sensors signify an increased chance of flash flood. The experiment was conducted on the seashore and it was observed that carbon dioxide level increased during wave run-up. To validate the wave run-up, a moisture sensor was utilised to measure the moisture content of the sand. Both of these readings from the sensors were taken into consideration to predict the flash flood. Furthermore, the multilayer perceptron (MLP) algorithm was trained to reduce the number of fake alarms [35].

There are several studies that show how to establish and harness a network of connected sensors for water level monitoring. Noar et al. [36] show how the Blynk platform can be utilised to connect the ultrasonic sensor with the internet and obtain real-time information on mobile phones. The proposed approach utilises NodeMCU as a medium to connect the range finder sensor with the Internet and receive information about the status of water level in real time. In a similar approach, Purkovic et al. [37] designed a low-cost ultrasonic sensor that was utilised along with other sensors from EnOcean. The data was transmitted every 5 min and the maximum range of the sensor was 10 m, with a resolution of 10 mm. However, the paper does not provide information about the results obtained from the experiment. Kafli et al. [38] proposed an IoT platform along with several sensors including rangefinder, humidity, carbon monoxide and a GPS sensor to monitor water level. The study was designed to be able to monitor water level in real time and issue early warnings to the local community. Chandanala et al. [39] proposed a technique to make the wireless system more energy efficient by optimising the parameters of network coding and duty cycling. Flooding was predicted by executing active monitoring through available off-the-shelf sensors such as an ultrasonic/sonar for estimating the water level and a precipitation sensor for estimating the intensity of rainfall. 

Furthermore, an early flood detection system can be implemented through real-time monitoring of the flood-prone area via sensors deployed in optimal locations at the site. This approach provides a convenient and cost-effective way to monitor flood-prone sites in real time [40]. Furthermore, Thekkil et al. [41] and Balaji et al. [42] utilised ZigBee and Global System for Mobile (GSM) to transmit acquired camera images and generate flood-related warnings. The study also utilised the scale-invariant feature transform (SIFT) algorithm for the autonomous monitoring of flood. In a similar approach, Pratama et al. [43] utilised Mamdani fuzzy logic along with ZigBee and water level sensors to detect and transmit the flood-related data. The study suggests that the maximum error for the proposed approach falls within an acceptable range of five percent. Waleed et al. [44] proposed a microchip-based solution using an array of piezoelectric pressure sensors that measure the pressure exerted by water and ZigBee for transmitting and receiving the data. The sensors were prototyped on Altera’s Cyclone board. The study also suggested that placement of the sensors is of extreme importance to forecast flood accurately. Ogie et al. [45] proposed a solution for the best placement of water-level sensors. The study puts a considerable emphasis on the optimal placement of the sensors, as it is important to gain situational awareness of water level in a large area of interest. The NSGA-II algorithm, which has gained wide application in many real-world problems, was used to find the best spot for the sensors. Using the sensor placement algorithm, four locally fabricated sensors were deployed to monitor water levels at different points in the waterways in Jakarta, Indonesia. In situations where accessibility is constrained, drones can be utilised to deploy sensors. For example, Abdelkader et al. [46] utilised UAVs to deploy cheap disposable sensors that can transmit data to UAVs about the monitored lake/valley.

Monitoring of water level has stirred the design and implementation of several wireless sensor networks (WSNs). For example, Wen-Yao et al. [47] utilised water level sensors along with analogue-to-digital converter (ADC) and an 8051 microprocessor in a ZigBee WSN to estimate water level. The study was executed to monitor and control the distribution substation in low-lying areas, providing early warning to the local community in case the water level increases above a predefined threshold value. Other similar studies have provided real-time signals from a WSN to inform an early warning system [48,49,50]. These studies have mostly relied on a web server to visualise the data coming from the flood monitoring station. Additionally, Jayashree et al. [51] proposed an early warning system based on real-time monitoring of dams via flow and water level sensors. The data collected from sensors is accessible and available to the public and can be fetched through an Android app designed for the research. Similarly, Teixidó et al. [52] and Smith et al. [53] presented a WSN system to notify the user in case of flooding. Similarly, Yumang et al. [54] designed a sensor network system capable of issuing warnings to locals in the event of flooding. The proposed system is based on sensors to monitor water level, a renewable power source to power the system and a GSM shield to transmit data. 

Data from sensor networks need to be validated and machine learning techniques can be quite useful in this regard. Machine learning techniques can be used in conjunction with ultrasonic/rangefinder sensors to predict flooding probability as needed for early warning [55]. Widiasari et al. [56] utilised the machine learning technique, Multilayer Perceptron (MLP), to analyse the time-series data coming from ultrasonic and precipitation sensors. The study was conducted to increase the accuracy of predicting flood events and also attributing floods in the region. Khan et al. [57] proposed an AI-based multi-modal network to alert locals to any upcoming flood event. The proposed approach is based on the sensor network, which consists of rangefinder, pressure, temperature, and gas sensors. The study indicated that the proposed system delivers accurate results with minimal false alerts. It would have helped to investigate the performance of the system in many locations. In a different study, Cruz et al. [58] developed a system to collect data from sensors such as a rain gauge, water level sensor, and soil moisture sensor. Using an artificial neural network (ANN) technique, real-time data from the flood monitoring station can be analysed to inform flood risk. The novelty in the study was the introduction of measuring river slope through the rangefinder sensor. The same authors in [59], Mousa et al. progressed their work further in [60] by introducing L1 regularization for fault detection and missing data points in real-time sensor applications. The proposed study also utilised ANN to compensate for the change in the environmental condition, accounting for how such change affects the readings obtained from sensors. In this ANN approach, the readings from multiple temperature sensors was obtained and the temperature variation between the ultrasonic sensor and ground determined in order to compensate for the error. The study highlights the fact that acquired data from sensors may not always be reliable as sensors may be damaged or covered with dirt; thus, early warning monitoring systems can issue false alarms.

The need to overcome the problem of false alarms has been of interest to several researchers. For example, Ancona et al. [61] proposed a technique that comprises intelligent sensors and 3D map techniques to forecast flooding while minimising false alarms. Horita et al. [62] validated WSN data about flooding with data reported by the citizens. In most cases, the sensors either were out of order or were not able to take the measurement. In a similar approach, Neal et al. [63] proposed a Kalman filter with WSN to improve the accuracy of the data coming from the sensors for flood forecasting. Ray et al. [64] discussed the IoT protocols utilized in the literature. Perumal et al. [65] proposed the IoT enable water monitoring station. Furthermore, Moreno et al. [66] and Purnomo et al. [67] proposed an early detection system by embedding rainfall, river slope and temperature sensors to monitor a continuous change in water level and forecast flash flooding. Moreover, Mostafa et al. [68] proposed a WSN along with a multi-agent system to classify whether the data coming from the sensors are valid or invalid. The study suggested an optimal model for aggregation and classification and is divided into three steps, namely, sensors verification phase, data aggregation and classification, and the database interaction step.

Review of the IoT-related literature presented above has revealed the potential of IoT-based sensors in early warning system. The most obvious findings to emerge from this subsection is that sensor-based approaches are more accurate in terms of calculating water level. However, the limitation of such approaches is that they only offer a reading at a single point and the only way to validate the reading is to visit the site due to the unavailability of visual data. Furthermore, we highlight the relevant studies that have focused on the IoT-based sensors as shown in Table 3.

### 3.3. Data Collection and Early Warning System

The accuracy of any deep learning computer vision application is dependent on the quality and quantity of the input dataset serving the neural network architecture for learning purposes [69]. Fuentes et al. [70] published the first image segmentation dataset of water body along with test results for three different CNNs. The deep learning approach on the image segmentation dataset proved to be very reliable, as the algorithm was able to learn on different images which reduced the need for manual filtering. The study concluded that Tiramisu image segmentation performed best on the water segmentation dataset. However, the dataset that the study utilised to train the model consisted of only 300 images with no benchmark available to compare. Understandably, data scarcity is a major limitation constraining AI initiative. For this reason, several studies have utilised crowdsourced social media images and textual data about flooding as a means of training and validating machine learning outputs in computer vision [71,72] In addition, Helber et al. [73] have proposed a dataset consisting of 27,000 geo-referenced labelled images which are divided into ten different classes. The collected images were from the Sentinel-2 satellite which opens up opportunities for a wide range of applications. The benchmark was created by using pre-existing CNN ResNet-50 architecture for the evaluation of the proposed dataset. The dataset and ground truths were collected manually to compare the accuracy of different algorithms. The ground truths were compared with predicted labels to evaluate the accuracy. A confusion matrix [73] was utilised to evaluate the performance of the proposed algorithm. A confusion matrix is a table that is often used to report the performance of a classification model on data for which ground truth values are known. The study compared the accuracy obtained from ResNet-50 with GoogLeNet CNN architecture, where ResNet-50 beat the GoogLeNet by 0.39%. The ResNet-50 achieved an overall classification accuracy of 98.57% in an RGB band combination.

Furthermore, the image data for computer vision can be collected through various means including ground cameras and UAV. Unmanned Aerial Vehicles are known to provide a fast and cost-effective approach to collecting data [74]. For example, Sullivan et al. [75] effectively utilised drones to collect stereo images of streambeds to gather information about the potential threat imposed by large woody debris (LWD) to culverts and bridges. Mourato et al. [76] explored the potential of using digital surface models (DSMs) generated from UAV-acquired RGB images as means of achieving optimised digital surface runoff models (DSRMs) which can then be inputted into hydraulic models to reduce spatial data uncertainties that often undermine the accuracy of flood hazard mapping. This entailed the filtering and removal of objects (e.g., buildings, trees and other vegetation) in order to obtain the digital terrain model (DTM) and a normalised digital surface model (nDSM) containing the height values of the objects. The GPS information was also added into the model to increase its accuracy. The study endorsed the concept of using UAVs for collecting data points from riverbed and terrain surfaces. The downside, though, is that the vegetation was not filtered out properly. This issue can be curtailed by using DSRM acquired through LiDAR technology. Moreover, Wang et al. [77] have proposed a multiple kernel fuzzy C means-Markov random field (MKFCM-MRF) model for the clustering of images obtained from UAVs. The advantage of using the MKFCM model is the reduction in noise while keeping the edge detection information preserved and the automatic optimisation of the eigenvector distribution in space. Researchers are using UAVs and sensors widely for applications involving monitoring of water levels, analysis and flood forecasting, as they provide flexibility, high spatial accuracy and a high sampling frequency rate. Furthermore, the collection of images in the 3D domain provides a better understanding of the site under investigation [78].

For computer vision applications, data plays an important role in the training of the algorithm. The study has found that the performance of an algorithm is directly proportional to the input data. Therefore, to make algorithms perform better in real-world scenarios, it is essential to train and test the algorithm on real-world images/data. Overall, this subsection reinforces the idea of collecting data for computer vision applications in flood monitoring and forecasting.

## 4. Computer Vision for Flood Modelling and Mapping

This section presents a review of relevant literature in terms of flood modelling and mapping. Overall, this section is divided into four subsections: overview of research progress; computer vision and data fusion for flood mapping; computer vision for debris flow estimation; and computer vision in estimating surface water velocity for hydrodynamic modelling of flood.

### 4.1. Overview of Research Progress

Flood mapping of large areas has benefited from the development in remote sensing technology [79], and the ability to extract water surfaces remains essential for flood-extent mapping [80]. The convenient way of acquiring images remotely of any location is through satellites [81]. Horkaew et al. [82] employed a cost-effective technique which is based on multivariate mutual information (MMI) and fused the acquired medium spatial resolution image from Landsat with a digital surface model (DSM). The reason for the fusion was to introduce topographic attributes to each coinciding pixel index of an image. The study concluded that the accuracy of the flood extent mapping was increased due to the context-based classification of an image. In a similar approach, Li et al. [83] developed software to automatically create near real-time flood mapping for the images retrieved from satellites. The proposed software can segment out the water body, cloud shadows, and terrain shadow from an image. However, the software is only limited to the USA and can cover any land region between 80 degrees south and 80 degrees north. 

Moreover, Martinis et al. [75] presented a two-phase flood monitoring system. First, the flood data is collected through moderate resolution imaging spectroradiometer (MODIS) and then it activates the second phase of the crisis management component which includes acquiring a large amount of spatial data from the satellite utilizing synthetic aperture radar (SAR). The study mentioned that TerraSAR-X-based flood mapping service could be triggered to derive high-resolution information for inundation mapping. Furthermore, the flood mapping accuracy can be increased by fusing the weak classifiers with the adaboost algorithm [84]. Liu et al. [85] proposed a novel approach to combine modest adaboost with the spatiotemporal context in order to increase the inundation flood mapping accuracy in the images obtained from satellites. The proposed approach takes the confidence value of each pixel into account so that it can find the pixels, which have a high probability for the training of the modest adaboost classifier. To monitor wetland areas in an arid Saharan region, Hakdaoui et al. [86] collected images from seven satellites (optical and radar) before and after flash flooding. The proposed approach is based on the combination of both spectral and categorical processing to obtain a resultant map of changes. In this sense, spectral indices (e.g., the albedo, NDWI, NDMI, and NMDI) were derived from multi-temporal optical remote sensing imagery and used to show where radiometric changes have occurred, whereas the categorical processing highlighted the thematic changes. The results demonstrate a robust methodology for determining the size of the area that is directly affected by a flash flood, further demonstrating that SAR images can complement optical images in flood mapping initiatives. This is an important achievement for desert wetland monitoring.

For improved efficiency in monitoring and flood mapping, it is vital to follow a UAV-routing strategy that maximizes area coverage. Malandrino et al. [87] proposed optimized route planning to achieve maximum coverage in applications using UAVs for emergency scenarios. The study aimed to determine the best coverage route that outputs the maximum user throughput across different regions of the topology. Furthermore, Popescu et al. [88] proposed a segmentation algorithm along with a flight plan for the flooded affected place. The study introduced a novelty in both flight planning for UAVs and the classification of the flooded area. The novelty in segmenting the flooded area derives from introducing colour information in the texture analysis of an image. For the proposed algorithm, the features of the images were selected using fractal techniques. The results showed that the introduction of such techniques helped to increase the accuracy of detecting flood up to 99.12%.

The high accuracy associated with UAV-based monitoring has helped a great deal in minimizing flood risk. For example, Casella et al. [89] utilized UAVs to monitor sea storms and their impact on coastal areas. The experimental results showed that the proposed approach of UAV photogrammetry and GIS offers cheaper and faster information without compromising accuracy. In a similar approach, Beni et al. [90] aimed to extract the water surface from images taken by UAVs [91]. The DEM was generated by utilizing the data points collected via the UAV. The data was then compared with the LiDAR sensor data from a satellite. The study found that data collected from UAVs are more accurate than LiDAR sensor data with an approximately 30 cm difference between the models. 

There is also research progress in classifying water surface from Landsat images. Landsat provides open-source data, but it suffers from low resolution. Isikdogan et al. [92] proposed an algorithm to segment out the surface water from land, clouds, ice, snow and shadows by using only Landsat band as an input. Currently, classification of the water surface from Landsat images suffers from false positives. This situation arises mainly due to the presence of cloud and terrain shadows, other reasons may include ice and snow threshold variations for different regions. The classification model takes the context of an image into account during the classification of an image. The proposed approach emphasized that the DeepWaterMap classification model works well across different terrain types and changing atmospheric conditions. The comparison between different models (conventional MLP and DeepWaterMap with one, three and five CNN layer blocks) can be seen in Table 4.

Moreover, Kang et al. [93] introduced an FCN-16 model based on fully convolutional networks (FCNs) for the mapping of flood events. The proposed approach achieved an improvement over FCNs to the overall accuracy of 0.0015 to 0.0058 under different test environments. The comparison between FCNs and FCN-16 can be seen in Table 5. Furthermore, in a study by Gebrehiwot et al. [94], the pre-trained FCN-16 model was further trained to extract flooded areas from UAV imagery. The FCN-16 model achieved an accuracy of 95% as compared to the 87.4% accuracy obtained with support vector machine (SVM). The confusion matrix was used to analyse the performance of the algorithm [94].

Instead of utilizing a satellite-based approach, wall-mounted cameras can be utilized for mapping of the flooded areas [95]. Lo et al. [96] introduced an image-based early warning system to instantaneously monitor and map a flooded area. This utilizes the existing video surveillance system and image processing techniques. The proposed method overcomes the need for a “staff gauge” or ruler to measure the water level. In this approach, the GrowCut method for region segmentation of an image was relied on to map the flooded area. During segmentation, the boundary between background and foreground was determined by the addition of the cellular automata (CA) algorithm. In a similar approach to GrowCut, Horng et al. [97] proposed a mean-shift clustering algorithm and region growing image segmentation algorithm to identify flooded areas and calculate the flood risk associated with the rise in water level. The proposed approach works well, as the purpose of utilizing region growing at the top of mean-shift is to group the pixels into meaningful clusters and analyse the variation in the growing region by comparing with previous frames. On a slightly different approach, Narayanan et al. [98] utilized the feature matching scale invariant feature transform (SIFT) algorithm to find standard features among two pictures which belong to the same building, whereas one picture was taken before the flood, the other was taken after the flood. To improve the generalizability of the algorithm, this study can be repeated on multiple images and sites.

Admittedly, some sites that require frequent monitoring are harder to access, but UAVs can provide a cost-effective alternative approach for real-time monitoring. Images taken from UAVs can support the localization, detection, segmentation and modelling of the flood [99]. Feng et al. [100] utilized drones to survey urban land to predict flood events. The reason for choosing UAV over static cameras was the ease of data collection at different locations. The study proposed the approach of a hybrid method based on the combination of texture analysis and the random forest algorithm. The overall accuracy for the proposed solution at the kappa index of 0.746 was 87.3%. The study proved that the accuracy increased up to 11.2% due to the addition of the texture analysis of the images. Important highlights of this study include an emphasis on utilizing a UAV platform for the monitoring of complex urban landscapes as well as the use of object-based information analysis (OBIA) to further increase the accuracy. Similarly, Popescu et al. [101] proposed an approach based on the analysis of texture feature and sliding box method via UAVs. The input image was divided into sub-images and classified into two classes, i.e., flooded or not flooded. The proposed algorithm was evaluated on ten images and achieved an accuracy of 98.57%. Even though the evaluation of this method could have benefitted from the use of a larger number of images, this level of performance is considered outstanding. 

Similarly, Sumalan et al. [102] developed a classification algorithm to classify images taken from UAVs into three different classes, namely, grass, buildings and flooded area. The proposed study developed an algorithm which is based on a local binary pattern so that it can extend to red and green channels in the RGB domain and to the h channel of HSV. The UAV was utilized to collect images, and the dataset was grouped into three categories. The histograms of the different classes were grouped together so that the histogram of the new image was compared against one of the three groups to predict the class of the input image. Instead of using the histogram approach, deep learning can be utilized to classify the images and the videos collected from UAVs autonomously into disaster and non-disaster categories [103]. Kamilaris et al. [104] utilized a deep learning model based on visual geometry group (VGG) to establish if an image is to be categorized into a disaster or not. The training used 544 images containing different images, some of which are non-disaster type and the others relate to disasters such as fires, earthquakes, collapsed buildings, tsunamis and flooding. By employing data augmentation techniques on the small dataset, an accuracy of 91% was achieved, with a suggestion that this accuracy can reach 95% with a larger dataset. However, little is known about how the accuracy of VGG architecture compares with other existing state-of-the-art CNN architectures.

This subsection presented the critical analysis of the cited literature in terms of proposed technology and the type of experimental setup for mapping of the flood events. From the analysis, it can be observed that data for mapping of the flood events can be collected by utilising ground, spaceborne and airborne sources. Review of the cited literature indicates that there is no single/general approach that would always work, the performance of the chosen method is highly dependent on the application and visual sensor location.

### 4.2. Computer Vision and Data Fusion for Flood Mapping

The accuracy of predictive models can be increased by fusing the data from different sources [105]. Zoka et al. [106] focused on combining data coming from radar SAR and optical data to monitor the water stretch in a wetland area after a flood. The study used a combination of categorical and spectral approaches, where radiometric changes were observed from optical sensing imagery, and thematic changes were observed from categorical processing. Moreover, the study suggested that the proposed methodology can be used to manage water storage capacity, and the flood extent mapping accuracy can be increased by merging categorical and spectral processing. In a similar approach, Chaouch et al. [107] utilized satellite images from radar SAR and Landsat to improve coastal flood inundation mapping. Different images from the satellites and aerial view were fused along with a digital elevation model (DEM) to make the proposed method of mapping more accurate. The proposed algorithm achieved an accuracy of 83% with the authors emphasizing that the accuracy can be further improved by pre-processing the data (removing inherent speckle noise from images) and increasing the quantity of the dataset. In addition, Senthilnath et al. [108] implemented a computer simulation of hierarchical clustering approach along with multi-purpose sensors including SAR for analysing data during the flood, and Linear Imaging Self-Scanning III (LISS III) for analysing the area before the flood. The data from both the sensors were mapped together to obtain the flooded and non-flooded areas. There are also studies fusing sensor and satellite image data. For example, Khan et al. [109] developed an approach to monitor flooding using optical imaging and a water-level sensor to find the water level in the extreme rainfall season. The data acquired from sensors and satellites were fused together to increase the accuracy of the proposed approach. The study also emphasized that the efficiency of emergency services can be increased if they are informed fast enough with accurate data of any disaster-affected site.

Disaster-affected areas require faster and efficient coverage so that help can be provided to people adequately [110]. Balkaya et al. [111] emphasized the need for an infrastructure that can deliver real-time data about the disaster-affected zones so that damage estimation can be swift. The study noted that the most accurate solution will be the multi-viewpoint of image/video fusion based on data input from both satellites and ground stations. To develop a 3D model of the terrain, Langhammer et al. [112] utilized a combination of data sources, including UAV aerial imagery as well as the data from ultrasonic sensors, which served as the hydrological data for determining the water depth. The proposed approach fused the data from both sources to build a reliable and precise hydrodynamic model. The study reports that the proposed approach of collecting data and developing the hydrodynamic model is very cost-effective and it enables the rapid development of models in a dynamic environment, especially in remote areas where the conventional data collection coverage is not available. In a similar approach, Zhu et al. [113] discussed the incentives of using UAVs to collect photogrammetry images along with geographical information systems (GIS) data on the potentially flooded areas. The study proposed an algorithm to create a flight plan for UAVs and was tested to monitor flooded areas in the urban region containing large buildings. The results showed that images collected along with GIS location points provided good insight into the flooded area as compared to conventional data collection techniques.

Review of the related literature presented in this subsection has demonstrated the potential of fusing data from two different data sources. Providing extra information to an algorithm helps it to perform better in real-world applications, where one source of data is an image and another source of data can be IoT-based sensor data, DEM, GIS, etc. The new information/data helps to improve the performance of the sensors working in a dynamic environment.

### 4.3. Computer Vision for Debris Flow Estimation 

Segmenting debris flow out of the running stream is one of the applications of computer vision [114]. Kao et al. [115] utilized spatial filtering techniques to monitor and detect debris flow in the running stream. They also utilized techniques such as background separation and entropy determination to overcome colour similarity and other non-ridge properties. Furthermore, they discussed the inclusion of luminance/chrominance (YUV) transforms, defining the ROI region that helps in improving accuracy for identifying debris flow. To improve the generalizability of the algorithm, it will be important to consider how to handle the definition of a threshold for the different filters which will change with time and place. Langhammer et al. [116] presented a novel approach to detect objects during flooding events through UAVs equipped with panchromatic cameras. The study proposed a workflow that uses a method of texture analysis, photogrammetric analysis and a classification model based on a 2D ortho-photograph and a 3D digital elevation model (DEM). The accuracy of the model depends upon the combination of image information (RGB, texture analysis, terrain ruggedness index (TRI) and DEM) that the model uses during evaluation. The comparison of the classification accuracy for the different combinations of input features can be seen in Table 6.

Flood debris detection, monitoring and accessing the damage due to the debris flow is an application of computer vision. This subsection highlighted how computer approaches can be used to detect and map the debris flow in a running stream. 

### 4.4. Computer Vision in Estimating Surface Water Velocity for Hydrodynamic Modelling of Floods

This subsection includes research where computer vision is used to estimate the flow rate and surface water velocity for hydrodynamic modelling. Finding the flow rate of water is of extreme importance in hydrological modelling and flood inundation mapping [117]. Optical flow is a method in computer vision that has been used to detect the movement of objects between two consecutive frames in a video sequence [118]. Harjoko et al. [119] successfully utilised the pyramidal Lucas–Kanade optical flow method for determining the flow rate of water in a case study of a dam. The directional arrows in the ROI is detected by the coordinates of the moving objects. Moving objects that have a directional vector parallel to the flow direction are useful for calculating the flow velocity. Discharge is also considered useful for modelling the relationship between rainfall and flash floods [120]. Al-Mamari et al. [120] utilised the large-scale particle image velocimetry (LSPIV) and the space-time image velocimetry (STIV) techniques to model the river discharge and established the relationship between high-intensity rainfall and flash floods. The study concluded that the flow was two-dimensional and time varying. However, the direction of the flow pattern was still determined with reasonable accuracy.

In a similar approach, Fujita et al. [121] studied the impact of snow-melting on floods by measuring the velocity and direction of the water. The far infrared (FIR) camera was utilized along with STIV techniques to conduct this study. Comparisons were made among readily available sensors such as acoustic Doppler current profilers (ADCPs), radio-wave velocity meters and image processing techniques. The study emphasized the idea of using image techniques, as the error between ADCPs and FIR cameras is less than 10%. The suggested direction for future work includes examining the effect of rainfall and wind on the accuracy of STIV measurements.

## 5. Analysis of Computer Vision Against Addressed Requirements

This section summarizes the cited literature related to computer vision as discussed above in two separate tables, i.e., Table 7, Table 8 and Table 9 which address computer vision techniques against the addressed requirements of accuracy, generalization and the scope of study. The accuracy and generalization of the proposed methods are measured against the experiment setup and the results that authors obtained in their research.

## 6. Recommendations for Future Research

Having reviewed the literature, we propose some directions for future research to address key areas that have remained unexplored. In delivering an early warning system, this review has found that image processing techniques, such as the OSF-based adaptive thresholding proposed in Reference [25] and computer vision techniques, such as CNN architecture proposed in Reference [20], are great starting points for the estimation of water levels. Both techniques were well defined and worked well in their respective applications. However, in both techniques, the camera was dependent on a staff gauge. These techniques can be further optimized where the physical water gauge/scale can be removed and replaced with the highly optimized CNN architecture and coupled with water level sensors so that the water level at any given point can be found without the need for a gauge. In future research, the IoT-based water level sensor [36,55,59,60] data can be fused with the data obtained from the camera, allowing for the camera to be calibrated in real time [112]. The main challenges mentioned above in the review are addressed against possible solutions and are summarised in the Table 10.

As this review has shown, regular monitoring of flood-prone areas is a challenging task and a costly activity for local governments [132]. This review has focused on studies that explore computer vision or IoT-based sensors to monitor or map floods. The findings have covered water-level monitoring in different sites that are of interest to understanding flood risks including residential street areas, rivers, urban drainage networks, seas, dams, lakes, etc. However, there is still a lack of studies on computer vision applications for the monitoring and management of coastal lagoons. Similarly, IoT-based sensors have not been widely applied in lagoon monitoring. Coastal lagoons provide a variety of essential services that are exceptionally admired by society, including storm defence, boating, recreation, fishing, tourism and natural habitats for aquatic lives [133]. However, coastal lagoons also pose a significant flood risk to residential areas adjacent to the lagoon foreshore. This flood risk is heightened by intense rainfall that causes water to build-up behind the closed entrance at lagoons. Hence, in the following sub-section we provide some recommendations for adopting computer vision and IoT sensors to improve the monitoring of lagoon sites.

### Recommended Future Research of Computer Vision and IoT Sensors in Monitoring Coastal Lagoons

Typically, coastal lagoons or lakes alternate between being closed and open to the ocean, forming what is commonly referred to as intermittently closed and open lakes and lagoons (ICOLLs). These are characterized by a berm, formed from sand and sediments deposited by winds, tides and waves from the ocean. This berm helps to prevent further flow of ocean water into the lagoon, but rainfall can cause the lagoon to overflow and inundate low-lying residential development. Knowing when to dredge the berm is therefore crucial for effective flood management and this would require regular monitoring of the water level in the lagoon, the berm height, berm composition and permeability and any activity related to artificial opening of the sand berm entrance. Hence, we recommend that future research explore the adoption of existing technology and techniques in computer vision and/or IoT-based sensors to monitor ICOLLs including obtaining berm height, water level measurement and improving decisions on when to open/close a lagoon entrance. The linear regression technique presented in Reference [19] can serve as a starting point for finding berm height. This study estimated the water level by finding the upper and lower limits of the dike area [19]. The adaptive method of finding the dike area assumed the upper limit to be a straight line because of the noise and thresholding limits in the proposed approach. This approach can be optimized to find the berm height. Moreover, the coastal lagoon entrance can be segmented out from the water region by utilizing the inundation mapping techniques such as region growing [97] and CNN architecture [92]. These techniques are utilized to segment the water surface, whereas for segmenting out the lagoon area, the CNN can be retrained with the addition of one more class, i.e., lagoon entrance. For adding one more class, additional data needs to be collected which will require an understanding of a new feature map of an image. Hence, further research should be undertaken to investigate semantic segmentation.

Furthermore, future research should be undertaken to automate the control process of a lagoon entrance by incorporating remote sensing and computer vision techniques. This will allow relevant data to be collected and visualized to understand the impacts of a change in weather conditions on berm height. At present, berm height is understood to be the product of wave run-up and the height of a berm which continues to increase until reaching the maximum height of a wave run-up [134]. The wave run-up varies for every beach and is directly affected by several factors such as beach slope, period, wave height and weather conditions [134]. Future research can explore the numerical computation of berm height using a mathematical model derived from experimentation. A data-oriented approach could produce interesting findings that will let researchers generalize the findings from one site to another lagoon site which may be behaving differently under different environmental conditions. In monitoring lagoon water levels, it might be possible to utilize the data fusion approach where data from the sensor can be fused with data from the camera to reduce false positives in water level readings. The reason for data fusion is that water level derived from the camera can be adjusted according to the single point IoT-based physical sensor so that readings can be obtained at multiple points without having to deploy physical sensors at several locations. In other words, the field of view (FOV) of the camera would give multiple points in the image, and each point can be considered as one physical sensor. This will suggest an improvement to coastal monitoring which is currently done either manually or from images taken from space.

## 7. Conclusions

This paper presented a systematic review of the literature regarding computer vision and IoT-based sensors for flood monitoring and mapping. The review found that there are a wide range of applications that support computer vision techniques and the IoT-based sensor approach for improved monitoring and mapping of floods. Some of these applications include, but are not limited to, an early warning system, debris flow estimation, flood risk management, flood inundation mapping and surface water velocity. It was observed that computer vision is advantageous for covering a broader range, and each point in the field of view (FOV) can be considered as one sensor when it comes to finding water level, whereas IoT sensors are more accurate but can only deliver a point-based reading. Therefore, both computer vision and IoT sensors have shortcomings that can be addressed by complementary use through the fusing of data coming from two independent sources of information and, thus, can improve the accuracy of the flood monitoring stations. It can be concluded that IoT-based sensor networks are essential in real-time monitoring of flooding, as they provide instant information about water levels thereby helping the responsible authorities to understand the impact of heavy rainfall on the carrying capacity of waterways so that adequate strategies can be put in place including the need for proactive emergency evacuation. Importantly, this study has revealed a lack of research focused on exploring computer vision or IoT-based sensors for improving the monitoring and management of coastal lagoon sites. Hence, some recommendations were made to direct future research, particularly in relation to monitoring berm heights in coastal lagoons.

## Figures and Tables

**Figure 1 sensors-19-05012-f001:**
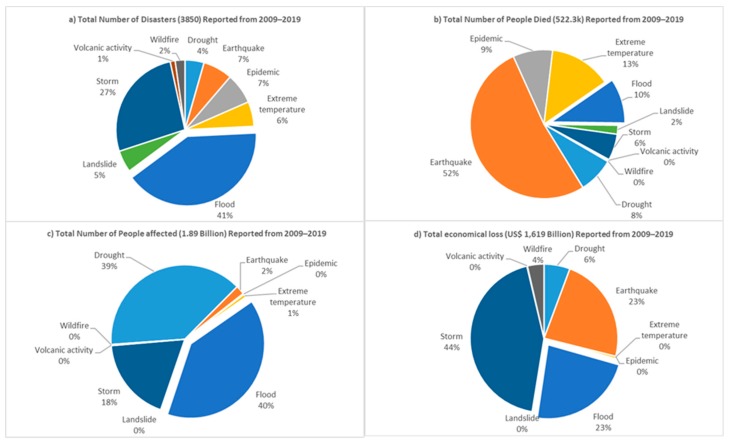
Comparison of different disaster types reported from 2009 to 2019: (**a**) total number of reported disasters; (**b**) total number of deaths; (**c**) total number of people affected; and (**d**) total economic loss [4].

**Figure 2 sensors-19-05012-f002:**
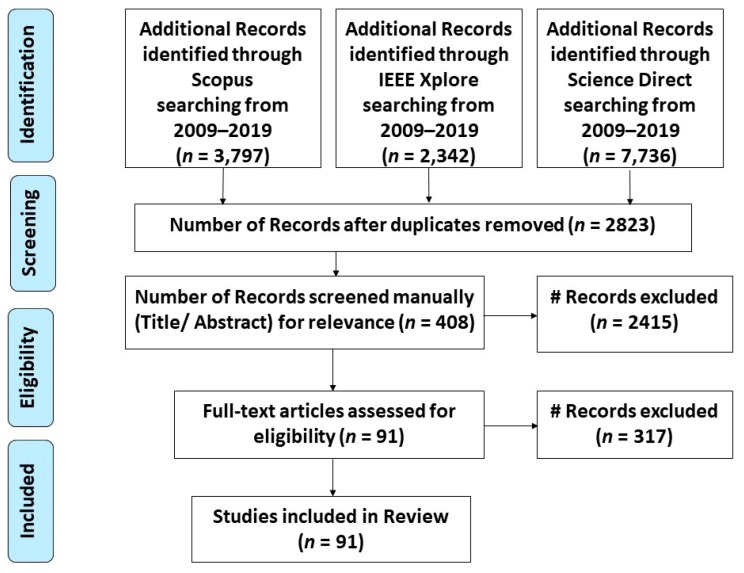
PRISMA flow diagram for the literature review [15].

**Table 1 sensors-19-05012-t001:** Keyword search results from different online scholarly databases.

Keyword	Scopus	IEEE Xplore	Science Direct
“remote sensing AND lagoon”	229	4	525
“remote sensing AND flood”	3022	871	452
“IoT AND flood”	48	58	921
“UAV AND flood”	109	36	521
“drones AND flood”	19	8	689
“computer vision AND flood”	40	58	792
“computer vision AND coastal”	30	52	1076
“wireless sensor network AND flood”	300	1255	2760
Total	3797	2342	7736

**Table 2 sensors-19-05012-t002:** Comparison of the average and variance of error for different computer vision techniques [20].

Method	Average Error (m)	Variance of Error (m^2^)
Difference Technique	0.046	0.003
Dictionary Learning	0.023	2.636 × 10^−4^
Convolutional Neural Network (CNN)	0.009	4.476 × 10^−5^

**Table 3 sensors-19-05012-t003:** Analysis of IoT-based sensors cited in the literature.

Purpose	Article	Proposed Method	Focus
A (sensors available to forecast flood)	[30]	Pressure transducer and radar sensor	Discussed the pros and cons of the pressure transducer and rangefinder sensors in estimating water level
	[34]	Optical and radar sensors	Comparison between optical and radar sensor for acquiring both time series and visual information
	[35]	Multilayer Perceptron (MLP) algorithm, along with soil moisture and CO_2_ sensors.	Forecasting of the flash flood by utilizing soil moisture and CO_2_ sensors
	[46]	Unmanned Aerial Vehicle (UAV) deployment of disposable sensors	One-time deployment of sensors to study the flow of river and forecast flooding
	[48,49,50]	Webserver for visualization of data	Forecasting of the flood by via remote sensing
	[64]	Internet of Things (IoT) protocols and commercial sensors	IoT for disaster management
B (IoT-based sensors and early warning system)	[36]	NodeMCU and ultrasonic sensor along with Blynk platform	Monitoring of water level in real-time via cell phone application powered by Blynk
	[37]	EnOcean and ultrasonic sensors	A cost-effective approach to deploy water level sensors
	[38]	Rangefinder, humidity, CO_2_ and Global Positioning System (GPS) sensors	Early warning system based on off-the-shelf sensors
	[39]	Precipitation and ultrasonic sensor along with utilizing network parameters to reduce power consumption	Power-efficient approach in WSN
	[41,42]	SIFT algorithm along with the camera, ZigBee and Global System For Mobile (GSM)	Early warning system based on ZigBee and GSM
	[43]	Mamdani fuzzy logic, ZigBee and water level sensor	Forecast flooding based on Fuzzy logic
	[44]	Piezoelectric pressure sensors, Altera’s Cyclone board and ZigBee	Early warning system based on ZigBee
	[45]	NSGS-II algorithm	Best spot for the WSN to get the best coverage of the site
	[47]	Water level sensor, Analog to Digital Converter (ADC), 8051 microprocessor and ZigBee to monitor the water level	Monitor and control of distribution substation in low-lying areas, and issue early warnings in case of water overflow
	[51]	Flow, water level and ZigBee	Early warning system based on real-time monitoring of dams
	[52,53]	Low-power wireless sensor network (WSN)	Early warning system based on WSN
	[54]	Water level sensor, Global System For Mobile (GSM) and renewable power source	Water level monitoring over cellular Communications
	[65]	Wireless sensor network (WSN)	Early warning system based on WSN
	[66]	IoT Device, GSM	Sensor for River water level monitoring over cellular communications
C (WSN and machine learning)	[56]	MLP to analyse time series data from ultrasonic sensor	Early warning system by utilizing the machine-learning technique
	[57]	AI-based multi-modal network system consists of rangefinder, pressure, temperature and gas sensors	Notify and issue warnings to locals in case of flooding
	[58]	Artificial Neural Network (ANN) along with soil moisture, rainfall and water level sensors	Early warning system based on WSN and ANN
	[59]	WSN consists of a rangefinder, water height elevation, rainfall and temperature sensors	Early warning system based on WSN
	[60]	Artificial Neural Network (ANN) along with ultrasonic and temperature sensor to validate data coming from sensors	Reduce fake alarms by monitoring temperature variations between the ultrasonic sensor and ground surface
	[67]	Water level sensor	Early warning system to reduce flood risk
D (validation of data from sensors)	[61]	Intelligent sensors and 3D mapping for segmentation	Reducing fake alarms by adding visual information with a water level sensor
	[62]	WSN and Geographical Information Systems (GIS)	Validation of data by comparing with the flood events reported by citizens
	[63]	Kalman filter and WSN	Validation of data coming from sensors
	[68]	WSN along with multi-agent system	Classification between valid and invalid data received from the sensor

**Table 4 sensors-19-05012-t004:** Comparison of models between the conventional MLP and DEEPWATERMAP with one, three and five convolutional blocks [92].

Model	Precision	Recall	Overall Measure (F1)
MLP	0.61	0.67	0.64
DeepWaterMap-1	0.81	0.94	0.87
DeepWaterMap-3	0.91	0.88	0.90
DeepWaterMap-5	0.92	0.87	0.90

**Table 5 sensors-19-05012-t005:** Comparison between fully convolutional networks (FCNs) and the proposed algorithm FCN-16 [93].

FCN	FCN-16	Advantages of FCN-16 Over FCNs
Kernel size = 7 × 7	Kernel size = 3 × 3	The smaller kernel size of the FCN-16 can be trained on fewer training samples in a shorter time.
L2 regular function	Dropout layers	Inclusion of dropout layers in the FCN-16 can prevent the model from overfitting.
FCNs utilise skip connections to fuse shallow layers, localise features and use global features. FCNs extract the features from a shallow layer and concatenate them with output of deep layers in the network	The structure of the fusion layer is changed, as for FCN-16, the input to the convolution layer is the addition of both deep layers and shallow layers	The advantage of using FCN-16 is that the model can extract new features using both global and local features.

**Table 6 sensors-19-05012-t006:** Classification accuracy for different classes [116].

Class	RGB	RGB + Textural Features	RGB + Textural Features + Terrain Ruggedness Index (TRI)	RGB + Textural Features + TRI + DEM
Fresh sand accumulation	93.9	95.1	95.4	95.9
Fresh gravel accumulation	80	83.7	86.7	95.7
Old gravel accumulation	75.5	76	76	93.2
Bank erosion	61.7	71.1	98.1	98.3

**Table 7 sensors-19-05012-t007:** Analysis of computer vision applications against addressed requirements: Part A.

Purpose	Article	Type of Information	Proposed Method	Addressed Requirements		
				+ -> Average ++ -> Good +++ -> State of the Art		
				Accuracy	Generalization	Scope of the Study
A (water level estimation/early warning system)	[18]	Static Ground Camera	Difference Method	+	+	Real-world, tested on one river
	[19]	Static Ground Camera	Logistic Regression and WSN	++	+++	Real-world, tested on thirteen rivers
	[20]	Static Ground Camera	CNN Architecture	++	++	Real-world, tested on six scenes
	[22]	Static Ground Camera	Image Texture features	+	+	Real-world, tested on one river
	[23]	Static Ground Camera	Accumulated Histogram and Bandpass Filter	+	Not Addressed	In-lab experiment
	[24]	Static Ground Camera	Edge Detector and Far Infrared (FIR) filter	+	Not Addressed	In-lab experiment
	[25]	Static Ground Near Infrared (NIR) Camera	OSF-based adaptive thresholding	+++	++	Real-world, tested on one river
	[27]	UAV Mounted Camera	Canny Filter thresholding	++	+	Real-world, tested on one DAM
	[28]	Static Ground IP Cameras	Image Texture-based segmentation	++	++	Real world, tested on one river
B (surface water velocity for hydrodynamic modelling)	[119]	Static Ground Camera	Pyramidal Lucas-Kanade optical flow method	++	++	Real-world, tested on one river
	[120]	Static Ground Camera	LSPIV and STIV techniques	+++	++	Real world, tested on one river
	[121]	Static Ground FIR Camera	STIV technique	+++	++	Real world, tested on one river

**Table 8 sensors-19-05012-t008:** Analysis of computer vision applications against addressed requirements: Part B.

Purpose	Article	Type of Information	Proposed Method	Addressed Requirements		
				+ -> Average ++ -> Good +++ -> State of the Art		
				Accuracy	Generalization	Scope of Study
C (flood-related data collection)	[70]	Static Ground Camera	Tiramisu image segmentation algorithm along with database	++	+++	Real-world, multiple locations
	[71,72]	Social Media	Flood image segmentation dataset	Not Addressed	Not Addressed	Real-world, multiple locations
	[73]	Spaceborne	ResNet-50 along with flood image database	+++	+++	Real world, multiple locations
	[76]	UAV Mounted Camera	Digital Terrain elevation (DTE) dataset Collection	Not Addressed	Not Addressed	Real-world, multiple locations
	[77]	UAV Mounted Camera	Fuzzy C-means model to cluster images and database collection	++	++	Real-world, multiple locations
	[122]	UAV Mounted Camera	Stereo images collection for floods	Not Addressed	Not Addressed	Real-world, multiple locations
D (flood risk management)	[89]	UAV Mounted Camera	Aerial images inspection with Geographical Information System (GIS) data points	++	++	Real-world, tested on coastal environment
	[90]	UAV Mounted Camera	Digital Elevation Model (DEM) data collection via UAVs	++	++	Real-world, tested on one site but can expand out to other sites
E (debris flow detection)	[100]	UAV Mounted Camera	Fusion of random forest and texture analysis	++	++	Real-world, multiple locations
	[115]	Static Ground Camera	Spatial filtering and luminance / chrominance (YUV) transforms	++	+	Real-world, tested on one site
	[116]	UAV panchromatic camera	Texture analysis and DEM	++	++	Real-world, tested on one site

**Table 9 sensors-19-05012-t009:** Analysis of computer vision applications against addressed requirements: Part C.

Purpose	Article	Type of Information	Proposed Method	Addressed Requirements		
				+ -> Average ++ -> Good +++ -> State of the Art		
				Accuracy	Generalization	Scope of the Study
F (flood detection and inundation mapping)	[75]	Spaceborne	Near real-time monitoring by triggering TerraSAR-X	++	+++	Real-world, multiple locations
	[82]	Spaceborne	Fusion of MMI with DSM	++	++	Real world, multiple locations
	[83]	Spaceborne	Image retrieval and classification software based on CNN	+++	+++	Real world, multiple locations
	[85]	Spaceborne	Modest adaboost and Spatiotemporal Context	++	++	Real world, multiple locations
	[86]	Spaceborne	Gaussian kernels and Support Vector Machine (SVM)	++	+++	Real world, multiple locations
	[87]	UAV	Optimized route planning for UAV	+	+	Real-world, UAVs path planning for flood monitoring
	[88]	UAV Mounted Camera	Texture analysis and fractal technique	++	+	Real-world, tested on big dataset
	[92]	Spaceborne	Convolutional Neural Network (CNN) architecture	++	++	Real world, multiple locations
	[93]	Spaceborne	FCN-16 CNN	++	++	Real world, Multiple locations
	[96]	Static Ground Camera	GrowCut method and Cellular automata (CA) algorithm	++	+	Real-world, tested on one river
	[97]	Static Ground Camera	Mean-shift and region growing	+	+	Real-world, tested on one river
	[98]	Static Ground Camera	SIFT algorithm	+	Not Addressed	In-lab experiment
	[101]	UAV Mounted Camera	Texture feature analysis	+	Not Addressed	Real-world, tested on ten images
	[102]	UAV Mounted Camera	Accumulated histogram and clustering images into a group	+	+	Real-world, multiple locations
	[104]	UAV Mounted Camera	VGG–CNN with a custom dense layer	++	+	Real-world, CNN trained on 444 images and tested on 100 images
	[106]	Spaceborne	Fusion of radar SAR and optical data	++	++	Real-world, Multiple locations
	[107]	Spaceborne	Fusion of Landsat images with DEM	++	++	Real-world, multiple locations
	[108]	Spaceborne	Hierarchical clustering approach	+	++	Real-world, multiple locations
	[109]	Spaceborne	Fusion of water-level sensor and satellites images	+	+	Real-world, tested on one site
	[111]	Spaceborne	Fusion of static ground cameras and satellite images	++	++	Real-world, multiple locations
	[112]	UAV and Ultrasonic sensor	Fusion of ultrasonic and DEM data collected from UAV to make a 3D model	++	++	Real-world, tested on one site but can expand out to other sites
	[113]	UAV Mounted Camera	Fusion of GIS and aerial photography	++	++	Real-world, tested in urban environment
	[123]	Social Media	Pre-trained CNN on ImageNet with the addition of meta-data analysis	++	++	Real-world, tested on real images posted online.
	[124,125,126,127,128,129,130]	Social Media	Fusion of contextual information with Image	++	+	Real-world, tested on real images posted online.
	[131]	Social Media	CNN architecture and meta-data analysis	++	+	Real-world, tested on real images posted online.
	[94]	UAV Mounted Camera	FCN-16 Architecture	+	++	Real world, tested on big dataset

**Table 10 sensors-19-05012-t010:** Main challenges addressed against possible solutions and future research.

Main Challenges	Possible Solutions/Future Research
Computer vision algorithm dependent on physical measuring scale such as a staff gauge for measuring water level [20,25]	An image can be converted from a 2D to 3D domain [78] and then the water level can be measured by utilizing advanced computer vision techniques [69]
Traditional image processing techniques work in a controlled environment. Environmental variations require image processing techniques such as thresholding [27,96,97] and custom filters [18,22,23,24,41]	In order to generalize, the model computer vision technique, such as deep leaning, can be used to work in the dynamic environment [20,92,93,104,123]
Limited site coverage [35,59,67]	Data fusion and remote sensing techniques can be used to fuse data from different sources [106,107,108,109,110,111,112,113]
Lack of open-source data to train computer vision algorithms [20,86,92,93]	Data can be collected and opened to train the proposed model [70,71,72,73,76,77,122]
Limited generalizability of the proposed solutions [7,18,23,24,98,101,102,115]	Instead of using image processing techniques, advanced convolutional neural networks can be used [93] Generalizability of the model can be assessed by utilizing real-world data for the testing phase [72]

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
