# Peer review of "Computer Vision and IoT-Based Sensors in Flood Monitoring and Mapping: A Systematic Review"

_sensors, 2019, doi:10.3390/s19225012_

Round 1
Reviewer 1 Report
The authors present a discussion on computer vision techniques and IoT sensor for real-time flood monitoring, flood modelling, mapping, and early-warning systems. In overall, the manuscript is well written and easy to follow. However, I have a few minor comments: - After reading the manuscript, I am unable to the find the discussion from the authors point of view at the end of each section and subsection. The takeaway messages are missing, otherwise the detailed content looks like a summarized section of recent advancement. - Before stating of the survey, it would be better to preset the motivation or main challenges involved in the flood modelling, mapping, and early-warning systems. - Moreover, what is the future roadmap? The current future directions are mainly very general, no such direct connection with the challenges mentioned in the previous sections. - - A few more diagram or table to summarize the main discussion would be good, e.g, table 7 is too long, it can be further divided into small parts, if possible.Author Response
We are grateful for the helpful feedback from the Reviewer that helped us to improve the quality of the manuscript. We carefully responded to all points and have modified the manuscript accordingly.
Please see the attachment

Reviewer 2 Report
The manuscript provides a review of the literature on the use of computer vision and the use of sensors in the Internet of Things (IoT) for flood monitoring and flood mapping. The review included 2,823 unique published articles selected from a search for appropriate key words in the databases of Scopus, IEEE Xplore, and Science Direct. Of these articles, 408 were manually selected for a full-text review based on relevance and publication date. The manuscript provides insight into current techniques and technology of computer vision used for flood monitoring and flood mapping and recommendations for additional research. The manuscript provides a strong recommendation for additional research in the area of monitoring the flooding of lagoons.
The review seems to be comprehensive and is well presented. My only significant concern is the use of the phrase "water level". The phrase is somewhat ambiguous in that it may convey to some readers that the vertical height of the water above some datum is being measured, rather than water extent. It is not clear to me how the phrase is being used throughout the manuscript. For instance, in Section 3.1 lines 120-137, does "water level" mean that the water height is being measured - or water extent? It do not know how water height can be measured with the Otsu threshold method using imagery from a camera. It seems that there must be a gauge in the image to distinguish a water height using a camera image. It should be clarified throughout the manuscript what is meant by "water level".
Other comments
Line 96 The sentence: "The research articles were manually screened by reading the title and abstract." is redundant with lines 98-99: "The titles and abstracts of these 2,823 articles were manually screened for relevance, resulting in the exclusion of 2145 records."
Lines 98-99 and Figure 2 Check the numbers. 2823-2145=678. It is not clear where the number 408 came from.
Line 111 "been" can be removed
Table 2 I believe the variance of error should be measured in meters and not square meters. Please check this.
Lines 221-223 Note that ESA now provides satellite radar data from Sentinel 1a and 1b at no charge for research.
Lines 226 and 239 Please clarify if and why CO and CO2 are used for water detection.
Line 252 GSM should be defined. In general, write out abbreviations the first time they are used.
Lines 345-348 Please describe what method was used to determine the "truth" to establish "an overall classification accuracy of 98.57%". If another method was used, one could ask what is the accuracy of that method, and why not use that method preferentially if it is used to establish "truth". Certainly the reader could look it up in the paper cited, but in general, when an accuracy of a product is cited, it would be helpful to report the method used to determine that accuracy.
Lines 427-429 granulometry relates to the measurement of the distribution of grain size. It should be more clear how this relates to " reducing flood risk in the future" or the topic of the paper, flood monitoring and mapping, if it is related.
Lines 437-438 cloud and terrain shadows are major sources of false positives from Landsat-like data.
Lines 450-452 again what is the accuracy assessment based on?
Line 519 Please clarify what is "cleansing" of a dataset?
570-571 Please clarify what is meant by "vector direction horizontal to the flow rate detection area"; is "perpendicular to the flow direction" meant?
Table 7 and 8: Please clarify what is meant by the different numbers of plus signs (+, ++, +++) in the tables.
633 "freshwater" is the wrong term here. Ocean water is salty, freshwater has a very low concentration of salt.
Author Response
We are grateful for the helpful feedback from the Reviewer that helped us to improve the quality of the manuscript. We carefully responded to all points and have modified the manuscript accordingly.
Please see the attachment
